# Prevalence of domestic violence against women in informal settlements in Mumbai, India: a cross-sectional survey

Nayreen Daruwalla,[1] Suman Kanougiya,[1] Apoorwa Gupta,[1] Lu Gram [ID] ,[2] David Osrin [ID] [2]

[1]Programme on Prevention of Violence Against Women and Children, Society for Nutrition, Education and Health Action (SNEHA), Mumbai, Maharashtra, India
[2]Institute for Global Health, University College London, London, UK

**Correspondence to**
Dr David Osrin;
d.osrin@ucl.ac.uk

## ABSTRACT

**Objectives** Domestic violence against women harms individuals, families, communities and society. Perpetrated by intimate partners or other family members, its overlapping forms include physical, sexual and emotional violence, control and neglect. We aimed to describe the prevalence of these forms of violence and their perpetrators in informal settlements in Mumbai.

**Design** Cross-sectional survey.

**Setting** Two large urban informal settlement areas.

**Participants** 5122 women aged 18–49 years.

**Primary and secondary outcome measures** Prevalence and perpetrators in the last year of physical, sexual and emotional domestic violence, coercive control and neglect. For each of these forms of violence, responses to questions about individual acts and composite estimates.

**Results** In the last year, 644 (13%) women had experienced physical domestic violence, 188 (4%) sexual violence and 963 (19%) emotional violence. Of ever-married women, 13% had experienced physical or sexual intimate partner violence in the last year. Most physical (87%) and sexual violence (99%) was done by partners, but emotional violence equally involved marital family members. All three forms of violence were more common if women were younger, in the lowest socioeconomic asset quintile or reported disability. 1816 women (35%) had experienced at least one instance of coercive control and 33% said that they were afraid of people in their home. 10% reported domestic neglect of their food, sleep, health or children's health.

**Conclusions** Domestic violence against women remains common in urban informal settlements. Physical and sexual violence were perpetrated mainly by intimate partners, but emotional violence was attributed equally to partners and marital family. More than one-third of women described controlling behaviours perpetrated by both intimate partners and marital family members. We emphasise the need to include the spectrum of perpetrators and forms of domestic violence—particularly emotional violence and coercive control—in data gathering.

**Trial registration number** ISRCTN84502355; Pre-results.

## Strengths and limitations of this study

► A large cross-sectional survey of women in informal settlements.
► The study included emotional violence, coercive control and neglect, as well as physical and sexual domestic violence against women.
► The study considered perpetrators of violence other than intimate partners.
► The cross-sectional nature of the study limited the possibility of causal inference.

Sustainable Development Goal (https://sdg-tracker.org/gender-equality). Domestic violence against women harms individuals, families, communities and society. For individuals, it causes injury or death,[1 2] reproductive health problems, harmful drug and alcohol use, anxiety, depression, post-traumatic stress disorder, self-harm and suicide.[1 3] For families, it leads to miscarriage, induced abortion, stillbirth, low birth weight, preterm delivery and further violence.[4 5] For communities and society, it leads to lack of agency, limited participation and lost economic productivity.[5 6]

Globally, 30% of women have survived physical or sexual violence by an intimate partner or sexual violence by a non-partner.[4] The precision of the wording of this statement illustrates the challenge of defining violence against women. The United Nations Declaration on the Elimination of Violence Against Women defines it as 'any act of gender-based violence that results in, or is likely to result in, physical, sexual or psychological harm or suffering to women, including threats of such acts, coercion or arbitrary deprivation of liberty, whether occurring in public or in private life'.[7] This definition allows for a range of perpetrators and a range of forms of violence. The perpetrator may be a husband or partner (intimate partner violence), a family or household member, or someone known

## BACKGROUND

The global burden of domestic violence is underlined by its inclusion as an indicator of gender equality for the fifth United Nations

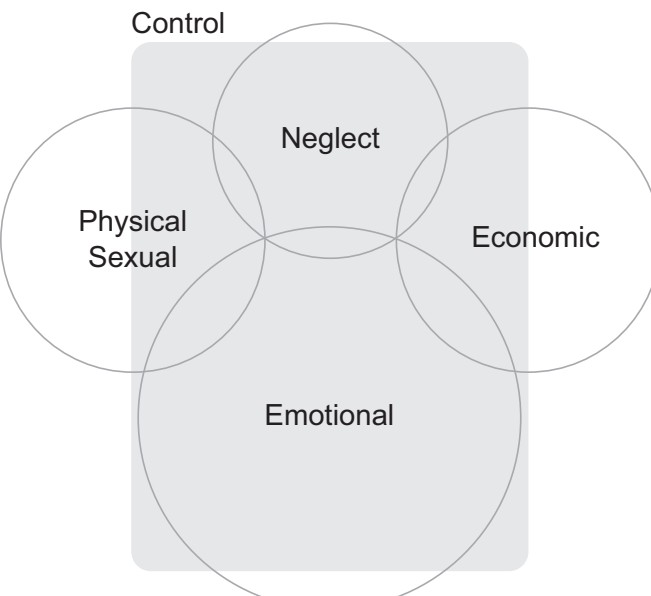

**Figure 1** Forms of domestic violence against women.

or unknown outside the domestic environment (in this paper, we define domestic violence as falling into either or both of the first two categories). Figure 1 conceptualises the intersecting forms of domestic violence: physical, sexual, emotional or psychological, control, neglect, or economic. These forms of violence overlap and occur in combination, in some settings with forms such as female genital mutilation and modern slavery.

There is consensus that measuring the scope and prevalence of violence against women is important, and that we should try to measure its different sources and forms.[8] Doing so may help identify the harm it can cause, advocate on behalf of individual women and against an unacceptable societal burden of coercive control (a perspective that owes much to the work of feminists over the last half century) and compare research findings across time and place and in a shared language.[9] We should be aware, however, that an emphasis on definition and metrics is characteristic of a public health approach.[10] As feminists have emphasised, violence against women is equally a human rights and criminal justice concern,[11] the product of an ecology of micro, meso and macro factors[12] and a manifestation of structural violence and gendered power imbalances. Too tight a focus on its epidemiology might depoliticise violence against women and shift attention away from these issues.[13]

Surveys have been important in international efforts to establish prevalence, the most prominent being the WHO multicountry study on women's health and domestic violence against women[14] and the International Violence Against Women Survey.[15] Choosing and comparing between the available scales and questionnaires is difficult.[16 17] The challenges to measurement are substantial,[18] not least in terms of definitions and terminology.[19] Early surveys focused on intimate partner physical violence. The Conflict Tactics Scale,[20] on which

they have often been based, has been extended to include sexual and emotional violence,[21] and the definition has been extended to include perpetrators other than intimate partners.[22] Whatever has been achieved in terms of definitions and scales, a fundamental challenge remains disclosure: women may not perceive, acknowledge or communicate that they are surviving violence.[23 24] This is a central issue for clinical and social service provision,[25–29] makes crime and hospital records difficult to use in estimating prevalence and extends to interviews in surveys.

India is a signatory to the Convention on the Elimination of All Forms of Discrimination Against Women.[30] Violence against women is addressed by criminal law, particularly domestic violence in Section 498-A of the Indian Penal Code, and by civil law in the form of the Protection of Women from Domestic Violence Act, 2005, which encompasses the varied forms of domestic violence and its perpetrators (available at https://indiacode.nic.in/bitstream/123456789/2021/1/200543.pdf). A systematic review of 137 studies suggested that 22% of women in India had survived physical violence in the past year (131 studies), 22% had suffered emotional or verbal psychological violence (60 studies), 7% sexual violence (79 studies) and 30% multiple forms of violence.[31] Women in India also commonly experience forms of violence that extend to coercive control and neglect,[32] articulated in the idea of gender-based household maltreatment[33] and captured in the recent Indian Family Violence and Control Scale.[34]

We work on primary, secondary and tertiary prevention of violence against women in informal settlements in Mumbai. The prevalence of physical and emotional violence in such settings has been reported as high.[35–38] At the beginning of a programme of community mobilisation to prevent violence against women and girls, we conducted a baseline survey. Our objectives in this analysis were to estimate the prevalence of physical, sexual and emotional violence, control and neglect. We were particularly interested in sources of domestic violence other than intimate partners and in forms of violence other than physical.

## METHODS
### Setting
One in four of the world's urban residents live in informal settlements (slums),[39] characterised by overcrowding, insubstantial housing, insufficient water and sanitation, lack of tenure and hazardous location.[40] In India, the state of Maharashtra accounts for 18% of the national total: more than 100 million residents.[41] Just over 40% of Mumbai homes are in informal settlements. The Census of India describes these as areas in which dwellings are unfit for human habitation as a result of dilapidation, overcrowding and poor building and street design, with associated poor ventilation, light and sanitation.[42]

The Society for Nutrition, Education and Health Action (SNEHA) is a non-government organisation working to improve health in such settlements. The

SNEHA programme on Prevention of Violence Against Women and Children began in 2000. Primary prevention is addressed through a combination of community group activities and resulting individual voluntarism. Secondary prevention includes local crisis response and psychological first aid by community organisers and referral to centres which provide counselling, legal and psychotherapeutic support, with links to the police and medical, shelter and social service providers. Tertiary prevention is provided primarily through referral to psychiatric and legal services. Having satisfied criteria for counselling, shelter, legal aid and access to medical care, SNEHA is a service provider under the Protection of Women from Domestic Violence Act, 2005, and is authorised to file domestic incident reports. Under the Protection of Children from Sexual Offences Act, 2012, SNEHA has reporting rights to the police and Child Welfare Committee.

### Design
We did a cross-sectional survey of women's experience of violence, collected at the beginning of a cluster randomised controlled trial of community interventions to prevent violence against women and girls (the SNEHA-TARA trial).[43]

### Sample size
We aimed to complete approximately 100 questionnaires in each of 50 clusters, giving a total sample of around 5000. An estimate of prevalence from a cross-sectional sample of 5000 in a population of 125 000 would have a precision of ~1%. Within this, a comparison of two categories of determinant for 100 respondents in each of 50 clusters would provide 80% power to detect a difference of 6% in prevalence estimates of 10%–20%.

### Sampling process
Potential clusters were identified in four phases in two areas of the city, and 50 were selected on the basis of (A) vulnerability assessed against a scorecard,[44] (B) no known plans for rehabilitation or demolition in the next 2 years, (C) absence of existing non-government organisation programmes addressing violence against women, (D) more than 75% of structures residential and less than 25% rental, and (E) clear separation from each other.

### Data collection
Structures and homes within clusters were mapped and 16 female interviewers with graduate education and 3 months of training visited households to enumerate residents and list possible participants. Interviewers began at a random start point and visited every second household to enrol participants. Inclusion criteria were that respondents should be women aged 18–49 years who were usual residents. When there was more than one potential respondent in a household, the investigator applied an algorithm that selected the youngest woman at risk of disability, followed by the youngest married woman, followed by the youngest unmarried woman.

The questionnaire included modules on general health and well-being, common mental disorder, household decision-making, household power and control, neglect, experience of economic, emotional, physical and sexual violence, disclosure and support. Questions were taken from existing Hindi versions where possible. If not, they were translated from English, piloted in two clusters external to the trial, amended and back translated.

Interviewers were all women and provided both time and sufficient information for women to consider whether to participate. They were supported by three field supervisors with direct linkage to counselling services, available by phone at any time. The interview team visited the local police station and social services to discuss their activities before starting in each area. Interviewers worked in groups of seven to eight in one cluster at a time, accompanied by a supervisor. A pair worked together in each lane and administered interviews in adjacent households.

To ensure privacy, interviews were arranged by advance appointment and avoided times when partners or children were likely to return from work or school. Women were interviewed at home or in a local community office if they preferred it. The interview began with general questions about demography, household residents, education, socioeconomic position, maternity and health. If a family member, neighbour or friend entered, the interviewer went back to asking questions about general health. If the person showed signs of staying, the interview was terminated and completed over up to three repeat visits. As a result of the gatekeeper consent process, community members were aware that interviewers would be visiting people in their area and this limited curiosity and intrusion.

### Data management
Interviewers used electronic tablets to enter information in a database in CommCare (www.dimagi.com). To optimise accuracy, the system included field constraints, lookup tables and automated skip logic. Mobile connectivity allowed immediate contact between interviewers and supervisors. We examined variation in prevalence rates by interviewer and discussed performance in supervisory meetings. We selected for field observation interviewers who showed signs of deviation from the group average and provided feedback where necessary.

### Statistical analysis
In developing our questionnaire, we aimed to provide data that would allow comparisons between studies and settings. We included questions from major surveys, international and national, and it should be possible to develop summary measures of violence against women using any of a number of combinations of questions. Our idea was to make the resulting anonymised data set available so that others might use it, choosing specific questions to facilitate comparison or pooling of data. Here we present composite indicators of lifetime and 12-month prevalence of physical, sexual and emotional violence.

Table 1 summarises the questions in three surveys from India (our TARA trial questionnaire, the National Family Health Survey (NFHS-4)[45] and the Indian Family Violence and Control Scale[34]) and two international surveys (the International Violence Against Women Survey[15] and the WHO multicountry study[14]).

Our composite estimate of prevalence of physical violence included nine questions that were comparable across surveys, although there was variation in whether threats of violence were considered incidents of physical violence. The estimate of prevalence of domestic sexual violence included four questions. The comparable questions about sexual violence were similar, although only the TARA survey and the Indian Family Violence Survey included a question on coercion to replicate activities seen in pornographic materials. The estimate of prevalence of domestic emotional violence included five questions that were comparable across surveys. We asked a series of questions about domestic coercive control, addressing mobility, education, employment, appearance and pregnancy. For control and neglect, we present the individual questions because composite indicators are not yet established.

We used the Washington Group Short Set on Functioning questions to identify disability, including difficulty with seeing, hearing, walking, remembering, self-care and speaking (http://www.washingtongroup-disability.com/washington-group-question-sets/short-set-of-disability-questions/). We took the most permissive of four possible approaches to classification, in which disability is registered if the respondent reports at least some difficulty in at least one of the six domains.[46] We classified socioeconomic position with quintiles of scores derived from standardised weights for the first component of a principal components analysis of 22 individual household assets.[47 48]

Prevalence is summarised by frequency and percentage. We plotted prevalence of physical, sexual and emotional violence in the last year by age group, schooling, socioeconomic asset quintile and disability, collapsed by grouping variable and rounded using the twoway (connected) command in Stata V.15. We calculated ORs for associations using logistic regression and Stata survey commands with cluster as the primary sampling unit and phase as stratum. We calculated intracluster correlation coefficients (ICC) for prevalence of physical, sexual and emotional violence in the last year using the loneway command in Stata V.15. The paper was prepared according to the Strengthening the Reporting of Observational Studies in Epidemiology statement for reports of observational studies (www.strobe-statement.org).

### Ethical considerations

We took permission for the survey from cluster gatekeepers identified by residents,[49 50] and followed WHO guidelines for research on domestic violence against women,[51] and on sexual violence.[52] Participants gave signed consent after discussing a participant information sheet. We made provision for storage of participant information sheets on women's behalf if they were concerned that the paperwork might be seen by others.

Data collectors explained their right to not provide answers to specific questions, to terminate the interview and to withdraw temporarily or permanently without penalty. All team members were trained in Good Clinical Practice for research ethics and participant protection (Scientia Clinical Services, 14 February 2019), and interviewers were supported by three field supervisors with direct linkage to counselling services.

Interviewing women about their possible experience of violence raises issues of consent, interviewer behaviour, privacy and confidentiality. Of particular concern is the duty of care after disclosure. We believe that an interviewee who discloses experience of violence should be offered optimal support, and interviewers were members of a broader team who were able to provide a full suite of crisis and counselling services, including home visits, medical, surgical and psychiatric referral, and negotiation with families, the police and legal representatives. When survivors disclosed violence, we followed established intervention protocols which included safety assessment, counselling, liaison with healthcare, police and legal services, and developing follow-up plans for the survivor and her family. Participants were able to speak with counsellors immediately by phone. When a survivor was not ready to disclose violence, the interviewer provided her with information on available services and legal rights and gave her a small card that was easy to hide and listed essential contact numbers and addresses for 24-hour crisis support, medical emergencies and the police. She took consent for any action from the participant herself.

### Patient and public involvement

Community members have been involved in the design of our research since 2000. It arises from the need to provide and improve services for survivors of violence, and a commitment to working with communities to prevent it. Research questions and outcome measures were derived from international efforts, but client priorities were reflected in some specific questions that we asked as a result of client experiences. All questions were discussed with participants in pilot exercises in order to understand their perceptions and ability to respond. Community guardians were involved in recruitment and the practicalities of administering the survey. Respondents took about an hour to answer the questions, and participated only after consideration. The general findings will be disseminated through community groups involved in ongoing interventions in the study areas.

### RESULTS
### Participants

Between 5 December 2017 and 28 March 2019, a total of 5277 households were approached for the survey. Four hundred and twenty-three (2%) had no eligible

**Table 1** Comparison of questions in five surveys describing physical, sexual and emotional violence

| | SNEHA-TARA | NFHS-4[45] | Indian Family Violence and Control Scale[34] | WHO multicountry study[14] | International Violence Against Women Survey[15] |
|---|---|---|---|---|---|
| **Physical** | | | | | |
| Pushed, shoved, shaken, hurt | √ | √ | √ | √ | √ |
| Twisted arm, banged head, pulled hair | √ | √ | √ | | √ |
| Slapped, pinched, bitten | √ | √ | √ | √ | √ |
| Hit, punched | √ | √ | √ | √ | |
| Kicked, dragged, beaten | √ | √ | √ | √ | |
| Things thrown at, burned | √ | | √ | √ | |
| Attacked or threatened with sharp object | √ | √ | √ | √ | √ |
| Attacked or threatened with blunt object | √ | √ | √ | √ | √ |
| Suffocated, choked, hung, poisoned | √ | √ | √ | √ | √ |
| **Sexual** | | | | | |
| Forced intercourse | √ | √ | √ | √ | √ |
| Forced other degrading act | √ | √ | √ | √ | |
| Threatened other act | √ | √ | | | |
| Forced to replicate pornography | √ | | √ | | |
| Forced sex during menses | √ | | √ | | |
| Forced sex with someone else | √ | | √ | | |
| Forced sex without contraception | √ | | √ | | |
| Threatened sex with another person | √ | | √ | | |
| Forced video of sex | √ | | √ | | |
| Taken advantage of when drugged or drunk | √ | | √ | | |
| Forced to watch pornography | √ | | | | |
| Insisted on repeated intercourse | √ | | | | |
| Forced to entertain others sexually | √ | | | | |
| Forced drugs or alcohol for sex | √ | | | | |
| Partner withheld sexual pleasure | √ | | | | |
| **Emotional** | | | | | |
| Insulted, made to feel bad about herself | √ | √ | √ | √ | |
| Belittled, humiliated in front of others | √ | √ | √ | √ | |
| Ignored, treated indifferently | √ | | √ | √ | |
| Scared or intimidated on purpose | √ | | √ | √ | |
| Threatened to hurt her, someone close, or take child away | √ | √ | | √ | √ |
| Insulted for not having a baby | √ | | √ | | |
| Insulted for not having a son | √ | | √ | | |

Continued

**Table 1** Continued

| | SNEHA-TARA | NFHS-4[45] | Indian Family Violence and Control Scale[34] | WHO multicountry study[14] | International Violence Against Women Survey[15] |
|---|---|---|---|---|---|
| Afraid of family members | √ | | | | |
| Accused of infidelity | √ | √ | | √ | |
| Insulted for being a woman | √ | | | | |

NFHS, National Family Health Survey; SNEHA, Society for Nutrition, Education and Health Action; TARA, Taking Action Reaching All.

female resident and in 544 (3%) an eligible woman was unavailable after three visits. Interviewers were unable to achieve privacy in 592 cases (11%) and 155 (3%) potential respondents declined the interview. The final survey included 5122 respondents. A median of 101 interviews were achieved in each cluster (IQR 100–103; range 94–118).

Table 2 summarises the characteristics of households, respondents and their partners. Dwellings were generally of robust or mixed robust and insubstantial construction (93%). Only 17% had their own toilet. Respondents were predominantly of Hindu (59%) or Muslim (37%) faith and 58% described themselves as of general caste. Sixty-eight per cent of women had been born outside Mumbai, 96% had been married or partnered, 2% had separated or divorced and 2% had been widowed. Thirty-six per cent had married before the age of 18 years, 88% of them by arrangement, 14% had had a pregnancy before the age of 18 years and 36% had three children or more. Women were a mean of 32 years old (SD 7.3) and their partners a mean of 36 (SD 8.1). Eighteen per cent of women had had no schooling, compared with 10% of partners, and 36% had completed high school, compared with 48% of partners. Only 24% of women had undertaken paid work in the last year, compared with 98% of men. For these women, the main types of employment were home based (62%), informal or formal services (17%) and domestic labour (10%). The main types of employment for men were in informal or formal services (49%) or as drivers (18%).

**Domestic physical violence**

Table 3 presents lifetime and 12-month prevalence of domestic violence. It includes individual questions and composite estimates for physical (nine questions), sexual (four questions) and emotional violence (five questions). Twenty-five per cent of women had ever experienced domestic physical violence and 13% had experienced it in the last year. This was usually intimate partner violence (11%), but 2% had experienced physical violence from their marital family and 2% from their natal family. Around 4% of women had been kicked, dragged or beaten in the last year, and around 3% had been threatened or attacked with a blunt or sharp object.

**Domestic sexual violence**

Six per cent of women had ever experienced domestic sexual violence and 4% had experienced it in the last year, almost all by their intimate partner. Removing the question about coercion to replicate pornography from the composite estimate reduced the estimated prevalence of sexual violence by 0.1% (domestic sexual violence to 5.4% ever (274 reports) and 3.5% in the last year (179), and intimate partner sexual violence to 5.1% ever (262) and 3.5% in the last year (177)). Additional questions suggested that 2.5% of partners had insisted on repeated intercourse (126 women in the last year), that 1.1% of wives had been forced to view pornography (56) that 0.8% had been subjected to coercive sex without contraception (39) and that 1.0% of partners were reported as withholding sexual pleasure from their wives (49). The prevalence of domestic physical or sexual violence in the last 12 months was 14% for currently married women, but 29% for women who had separated from their partner.

**Domestic emotional violence**

Nineteen per cent of women had experienced emotional violence in the last year. The source was roughly evenly split between intimate partner (12%) and marital family members (11%). Emotional violence came from both intimate partner and marital family for 5% of women. Six per cent of women had been accused of infidelity, 6% had been insulted for being a woman and 8% of 505 women with disability had been insulted for it.

**Summary figures**

Limiting the data set to 4913 ever-married women for comparison with surveys such as the NFHS, the prevalence of domestic physical violence in the last 12 months was 13%, of sexual violence 4%, of physical or sexual violence 14% and of emotional violence 19%. Recent guidance suggests that studies present the proportion of ever-partnered women and girls aged 15 years and older who have survived physical, sexual or psychological violence by an intimate partner in the last year.[53] For intimate partner violence (a subgroup within domestic violence) our findings were 11% for physical violence, 4% for sexual, 13% for emotional (psychological), 13% for physical or sexual and 17% for physical, sexual or emotional violence.

Figure 2 shows prevalence of physical, sexual and emotional violence for each of 50 clusters. The ranges

**Table 2** Characteristics of households, respondents and their partners

| Household | n | % |
|---|---|---|
| Fabric | | |
| Kachha (insubstantial) | 353 | 7 |
| Mixed | 2149 | 42 |
| Pukka (robust) | 2620 | 51 |
| Toilet | | |
| Private | 875 | 17 |
| Public or charity | 4245 | 83 |
| Open defecation | 2 | 0 |
| Religion | | |
| Hindu | 3002 | 59 |
| Muslim | 1902 | 37 |
| Buddhist | 158 | 3 |
| Other | 60 | 1 |
| All | 5122 | 100 |

| Individual | Female respondent | | Husband or partner | |
|---|---|---|---|---|
| | n | % | n | % |
| Age group (years) | | | | |
| 18–19 | 115 | 2 | 14 | 0 |
| 20–29 | 2038 | 40 | 920 | 19 |
| 30–39 | 2056 | 40 | 2108 | 44 |
| 40–49 | 913 | 18 | 1372 | 29 |
| 50+ | | | 391 | 8 |
| Education | | | | |
| None | 947 | 18 | 474 | 10 |
| Primary 1–5 years | 859 | 17 | 590 | 12 |
| Middle 6–8 years | 1122 | 22 | 981 | 20 |
| High 9–10 years | 1158 | 23 | 1543 | 32 |
| Senior 11–12 years | 572 | 11 | 698 | 15 |
| Undergraduate | 363 | 7 | 371 | 8 |
| Postgraduate | 100 | 2 | 123 | 3 |
| Other | 1 | 0 | 25 | 1 |
| Paid work in the last 12 months | | | | |
| Yes | 1252 | 24 | 4697 | 98 |
| No | 3870 | 76 | 108 | 2 |
| All | 5122 | 100 | 4805 | 100 |

were substantial: for physical violence 2%–27% (IQR 12%–16%), for sexual violence 0%–8% (IQR 2%–5%) and for emotional violence 7%–32% (IQR 15%–23%). ICCs were 0.023 for domestic physical violence in the last year (95% CI 0.011 to 0.036), 0.005 for sexual violence (95% CI 0.000 to 0.010) and 0.016 for emotional violence (95% CI 0.006 to 0.026).

Figure 3 summarises prevalence of physical, sexual and emotional violence by age group, schooling, socioeconomic asset quintile and disability. Women were less likely to face physical and emotional violence as they aged: the prevalence of physical violence halved between the 20s and the 40s. There was no obvious pattern for schooling. All three forms of violence were more common against women in the lowest socioeconomic asset quintile and against women with disability. The ORs in figure 3 did not change substantially when all four covariates were included in multivariable models, apart from disability, which was associated with an adjusted OR of 1.68 (95% CI 1.24 to 2.29) for physical violence, 2.18 (95% CI 1.37 to 3.47) for sexual violence and 1.66 (95% CI 1.31 to 2.12) for emotional violence.

### Domestic coercive control

Table 4 summarises forms and sources of coercive control. We took a conservative approach to classifying controlling behaviour. A restrictive behaviour was registered only if it was reported as always occurring, and a permissive behaviour only if it was reported as never occurring. Overall, 1816 women reported being subject to at least one controlling behaviour in the list in table 4 (35%), although we recorded responses to a larger set of questions. Respondents commonly reported that their mobility and socialising were restricted: 26% said that they always required permission to go out, 14% that they were never allowed out in the evening and 10% that they were always accompanied when they did go out. Forty-eight per cent were never allowed to meet with male friends, 10% were never allowed to meet with female friends and 5% were never allowed to meet their natal family. At home, 33% of women said that they were afraid of family members. Thirteen per cent were never free to speak on the phone and 10% were never able to speak freely in the home. These kinds of restrictions were described as coming more often from marital family members than from partners. In terms of sexual and reproductive health, 52 women had been denied access to contraception and 15 had been coerced to use it. Thirty-seven (1% of 4577) had been prevented from having a termination of pregnancy while 15 (0.3%) had been coerced to have one.

### Domestic neglect

Seven per cent of women reported that their family had neglected their food in the last year, 8% that they had neglected their sleep, 8% that they had neglected their health and 12% of 4375 with children that they had neglected their children's health. Collectively, 10% of women reported at least one of these four dimensions of neglect.

### Disclosure of violence

Of 1153 women who had survived physical, sexual or emotional domestic violence in the last year, 47% had disclosed to someone. The main confidants were family and friends (40%) and teachers, faith leaders or local

**Table 3** Lifetime and 12-month prevalence of domestic physical, sexual and emotional violence against 5122 women aged 18–49 years, by perpetrator: intimate partner, marital family or natal family member

| | Lifetime | | Lifetime by perpetrator | | Last 12 months | | Last 12 months by perpetrator | |
| --- | --- | --- | --- | --- | --- | --- | --- | --- |
| | Domestic | Intimate partner | Marital | Natal | Domestic | Intimate partner | Marital | Natal |
| | n (%) | n (%) | n (%) | n (%) | n (%) | n (%) | n (%) | n (%) |
| **Physical violence** | | | | | | | | |
| Pushed, shoved, shaken, hurt | 595 (11.6) | 458 (8.9) | 186 (3.6) | 33 (0.6) | 292 (5.7) | 247 (4.8) | 70 (1.4) | 17 (0.3) |
| Twisted arm, banged head, pulled hair | 464 (9.1) | 371 (7.2) | 121 (2.4) | 23 (0.4) | 224 (4.4) | 194 (3.8) | 43 (0.8) | 10 (0.2) |
| Slapped, pinched, bitten | 1154 (22.5) | 986 (19.3) | 134 (2.6) | 139 (2.7) | 565 (11.0) | 504 (9.8) | 52 (1.0) | 71 (1.4) |
| Hit, punched | 421 (8.2) | 351 (6.9) | 90 (1.8) | 28 (0.5) | 206 (4.0) | 182 (3.6) | 30 (0.6) | 10 (0.2) |
| Kicked, dragged, beaten | 398 (7.8) | 326 (6.4) | 88 (1.7) | 28 (0.5) | 199 (3.9) | 175 (3.4) | 32 (0.6) | 10 (0.2) |
| Things thrown at, burned | 51 (1.0) | 41 (0.8) | 14 (0.3) | 0 (0.0) | 15 (0.3) | 13 (0.3) | 3 (0.1) | 0 (0.0) |
| Attacked or threatened with sharp object | 96 (1.9) | 74 (1.4) | 22 (0.4) | 3 (0.1) | 51 (1.0) | 44 (0.9) | 6 (0.1) | 2 (0.0) |
| Attacked or threatened with blunt object | 218 (4.3) | 165 (3.2) | 51 (1.0) | 22 (0.4) | 103 (2.0) | 90 (1.8) | 16 (0.3) | 6 (0.1) |
| Suffocated, choked, hung, poisoned | 114 (2.2) | 96 (1.9) | 20 (0.4) | 5 (0.1) | 52 (1.0) | 48 (0.9) | 8 (0.2) | 1 (0.0) |
| **Sexual violence** | | | | | | | | |
| Forced intercourse | 231 (4.5) | 219 (4.3) | 6 (0.1) | 4 (0.1) | 138 (2.7) | 136 (2.7) | 1 (0.0) | 0 (0.0) |
| Forced other degrading act | 128 (2.5) | 126 (2.5) | 1 (0.0) | 1 (0.0) | 95 (1.9) | 95 (1.9) | 1 (0.0) | 0 (0.0) |
| Threatened other act | 74 (1.4) | 72 (1.4) | 0 (0.0) | 1 (0.0) | 43 (0.8) | 43 (0.8) | 0 (0.0) | 0 (0.0) |
| Forced to replicate pornography | 75 (1.5) | 75 (1.5) | 0 (0.0) | 0 (0.0) | 45 (0.9) | 45 (0.9) | 0 (0.0) | 0 (0.0) |
| **Emotional violence** | | | | | | | | |
| Insulted, made to feel bad about herself | 1296 (25.3) | 536 (10.5) | 877 (17.1) | 99 (1.9) | 690 (13.5) | 347 (6.8) | 419 (8.2) | 64 (1.2) |
| Ignored, treated indifferently | 773 (15.1) | 339 (6.6) | 541 (10.6) | 50 (1.0) | 449 (8.8) | 230 (4.5) | 290 (5.7) | 34 (0.7) |
| Belittled, humiliated in front of others | 682 (13.3) | 313 (6.1) | 445 (8.7) | 52 (1.0) | 417 (8.1) | 223 (4.4) | 242 (4.7) | 35 (0.7) |
| Scared or intimidated on purpose | 841 (16.4) | 592 (11.6) | 330 (6.4) | 67 (1.3) | 570 (11.1) | 432 (8.4) | 191 (3.7) | 52 (1.0) |
| Threatened to hurt her, someone close, or take child away | 213 (4.2) | 147 (2.9) | 100 (2.0) | 6 (0.1) | 130 (2.5) | 100 (2.0) | 49 (1.0) | 4 (0.1) |
| Any physical violence | 1293 (25.2) | 1084 (21.2) | 261 (5.1) | 170 (3.3) | 644 (12.6) | 562 (11.0) | 103 (2.0) | 84 (1.6) |
| Any sexual violence | 289 (5.6) | 277 (5.4) | 7 (0.1) | 4 (0.1) | 188 (3.7) | 186 (3.6) | 2 (0.0) | 0 (0.0) |
| Any emotional violence | 1599 (31.2) | 876 (17.1) | 1045 (20.4) | 138 (2.7) | 963 (18.8) | 607 (11.9) | 538 (10.5) | 90 (1.8) |
| All | 5122 (100.0) | 5122 (100.0) | 5122 (100.0) | 5122 (100.0) | 5122 (100.0) | 5122 (100.0) | 5122 (100.0) | 5122 (100.0) |

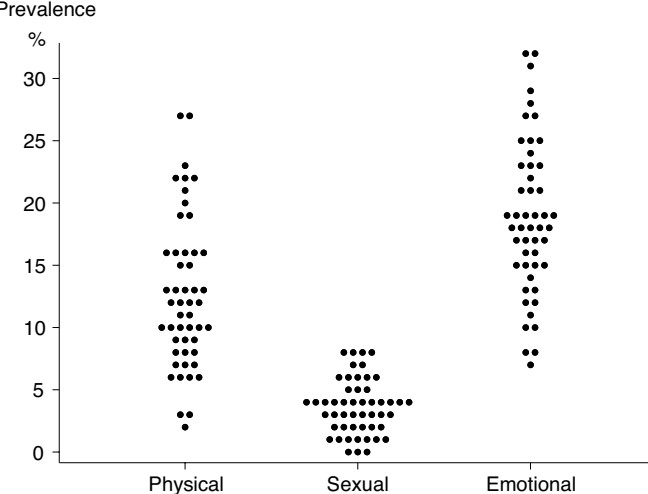

**Figure 2** Prevalence of physical, sexual and emotional violence against 5122 women aged 18–49 years in 50 informal settlement clusters in Mumbai.

leaders (4%). Even though 18% had sought medical help, only 8% had disclosed to a healthcare provider, social worker or counsellor. Twenty women approached our services for support within 2 months of the survey.

## DISCUSSION
### Key findings
Our cross-sectional survey of 5122 women aged 15–49 years in informal settlements in Mumbai documented high rates of domestic violence. In the preceding year, 13% of women had suffered physical violence, 4% sexual violence and 19% emotional violence. Physical and sexual violence were perpetrated mainly by intimate partners, but emotional violence was perpetrated equally by intimate partners and marital family members. Controlling behaviours were ascribed to both intimate partners and marital family members, with 35% of women reporting at least 1 of 12 restrictions.

### Limitations
Our findings have three general limitations. First, although the prevalence of domestic violence was high, we should assume some under-reporting given the constellation of forces inhibiting disclosure. Second, we preferentially recruited younger, married and disabled women within households, which would tend to increase reported prevalence. Third, our study was cross-sectional and subject to the usual caveats around causal inference and the accuracy of time recall.

### Comparison with other studies
We updated the review of quantitative studies by Kalokhe and colleagues,[31] identifying publications from 1 January 2015 to 1 November 2019 using similar search terms, and found eight more recent published studies. The reported prevalence of domestic violence was higher in rural[54–56] than in urban settings. The estimated lifetime prevalence

of physical violence ranged from 18%[57] to 27%,[58 59] of sexual violence from 10%[57] to 26%[59] and of emotional violence from 20%[57] to 43% (this high estimate included questions on economic violence).[59] A cross-sectional survey similar to ours, of 1137 mothers aged 18–39 years in an informal settlement in Mumbai, found a lifetime prevalence of physical intimate partner violence of 17%, of emotional violence of 12% and of sexual violence of 5%.[60]

Our findings also compare with the NFHS-4 for Maharashtra state, in which estimates of prevalence of intimate partner violence were based on interviews with 2472 ever-married women aged 15–49 years.[61] Both surveys estimated lifetime physical violence at 21%. The NFHS-4 estimate for the last 12 months was 15% (our study 11%). We found higher prevalence of sexual (5% ever compared with 2% in the NFHS-4; 4% in the last 12 months compared with 2%) and emotional violence (17% ever compared with 10% in the NFHS-4; 12% in the last 12 months compared with 7%). Our estimates of prevalence of all three types of violence were higher than those of the urban sample in the Maharashtra NFHS-4, which interviewed 1220 women (lifetime intimate partner physical violence 21% in our study, compared with 16% in the NFHS-4, sexual violence 5% compared with 2% and emotional violence 17% compared with 9%). There are three possible explanations for our higher estimates. First, they might be based on different sets of questions: we used similar questions and composites of them and do not think this was an issue. Second, they might reflect true population differences: our participants were all residents of informal settlements and might be more likely to face violence than the urban average. Third, women might have been more likely to disclose violence to our team of interviewers, who were backed up by a strong programme of support for survivors. It is also possible that women answer questions differently when they are responding to a survey on violence than when the questions on violence are included in a broad survey like the NFHS-4. In a consideration of methodological issues in research on intimate partner violence, Saltzman suggests that questionnaire context and cues given to the participant can influence response.[10]

### Implications
We have a clear picture of the high burden of domestic violence against women, and of the range of findings across urban and rural settings in India. The odds of physical violence are greater for younger married women, with poorer socioeconomic position, and where underage marriage is common.[62] We found that 47% of women who had suffered violence had sought help from somebody. This is considerably more than the NFHS-4, which found that 14% of urban women who had experienced physical or sexual violence had sought help from any source, usually natal family, marital family, friend or neighbour. Although disclosure to a healthcare provider, social worker or counsellor was uncommon (8%), it

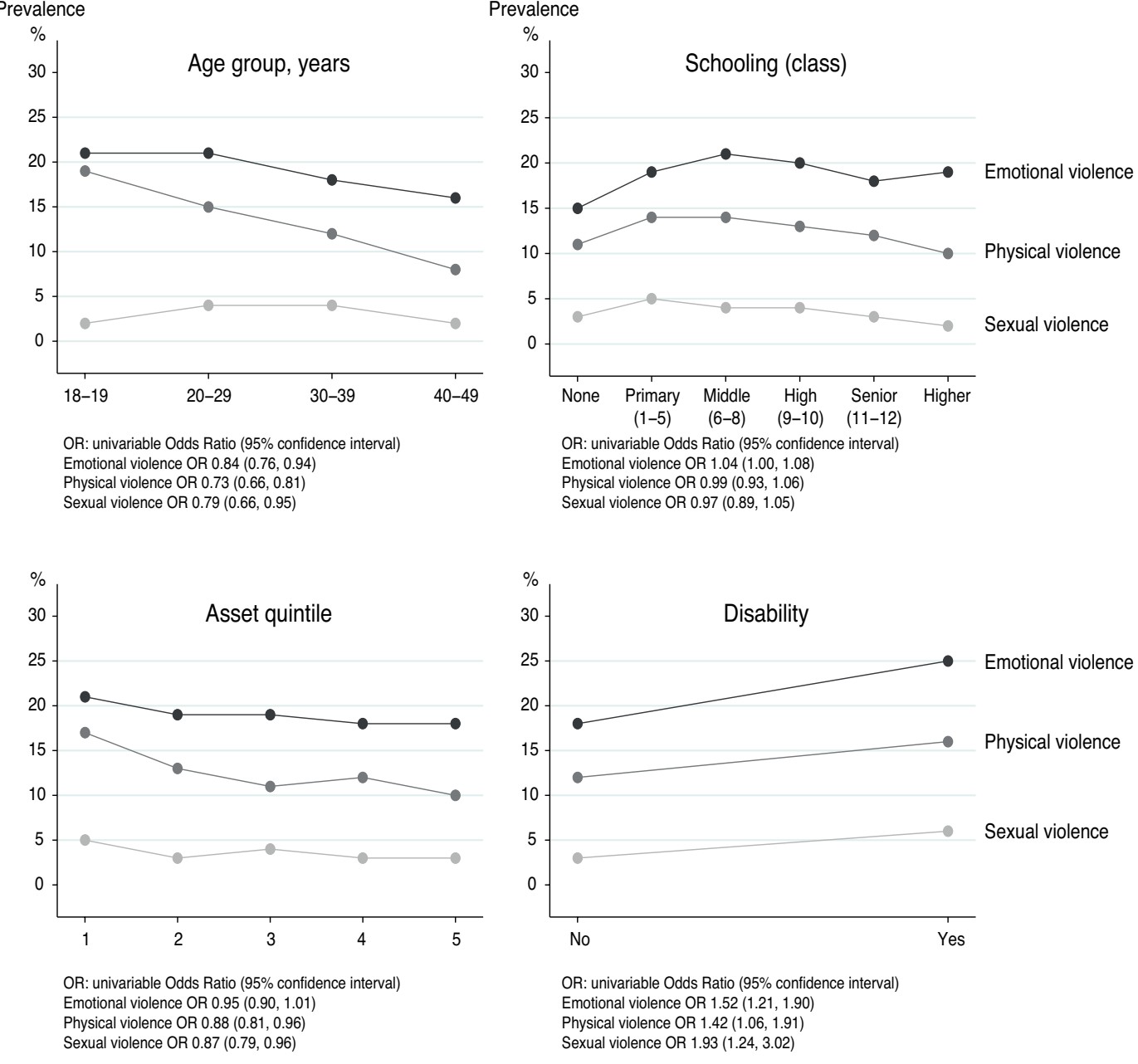

**Figure 3** Prevalence of physical, sexual and emotional violence against 5122 women aged 18–49 years, by age group, schooling, socioeconomic asset quintile and disability.

was also higher than the NFHS-4 urban figure of 3%. Perhaps this provides some hope. Awareness of domestic violence is certainly increasing and it may be that women are becoming more confident that what they are going through is violence and that it is not acceptable.

Our particular concerns are perpetration of violence by the marital family and the burden of emotional violence and coercive control. Although only 11% of studies in the recent review considered non-partner violence,[31] domestic violence by people other than intimate partners has long been a concern in the Indian context.[63] Women entering the marital household are traditionally subordinate and the spectrum of violence extends from subtle neglect to dowry death. Toleration of their subservience in exchange for security and later authority over their own daughters-in-law has been described as the patriarchal bargain,[64] and the burden of violence against women by mothers-in-law[65] and other family members has been termed gender-based household maltreatment. It informed the design of our own questionnaire[66] and includes control over reproduction, limiting contact with friends and family, and access to food.[32 34] These kinds of less visible emotional violence and control have been less well studied, but they need to be: marital family members were the source of emotional violence in about half of reports in our study and of 35% of domestic violence in rural Tamil Nadu.[54]

**Table 4** Lifetime prevalence of domestic control of 5122 women aged 18–49 years, by perpetrator: intimate partner, marital family or natal family member

| Control | Domestic | Intimate partner | Marital | Natal |
|---|---|---|---|---|
| | n (%) | n (%) | n (%) | n (%) |
| **Mobility** | | | | |
| Movement always monitored | 419 (8.2) | 278 (5.4) | 140 (2.7) | 44 (0.9) |
| Always prevented from attending meetings | 227 (4.4) | 129 (2.5) | 118 (2.3) | 5 (0.1) |
| **Education** | | | | |
| Prevented from schooling | 152 (3.0) | 53 (1.0) | 47 (0.9) | 62 (1.2) |
| **Employment** | | | | |
| Prevented from seeking employment | 853 (16.7) | 728 (14.2) | 162 (3.2) | 31 (0.6) |
| Coerced to seek employment | 78 (1.5) | 42 (0.8) | 35 (0.7) | 7 (0.1) |
| **Home dynamics** | | | | |
| Excluded from family matters | 366 (7.1) | 108 (2.1) | 300 (5.9) | 14 (0.3) |
| Dress or hairstyle always controlled | 267 (5.2) | 119 (2.3) | 145 (2.8) | 25 (0.5) |
| Limited access to house | 173 (3.4) | 56 (1.1) | 137 (2.7) | 7 (0.1) |
| Forced out of house | 165 (3.2) | 52 (1.0) | 121 (2.4) | 5 (0.1) |
| Locked in house | 39 (0.8) | 18 (0.4) | 22 (0.4) | 1 (0.0) |
| **Health** | | | | |
| Made to do excessive work | 239 (4.7) | 39 (0.8) | 209 (4.1) | 11 (0.2) |
| Always requires permission for healthcare | 753 (14.7) | 513 (10.0) | 286 (5.6) | 48 (0.9) |
| Any of the above | 1816 (35.4) | 1303 (25.4) | 830 (16.2) | 184 (3.6) |
| All | 5122 (100.0) | 5122 (100.0) | 5122 (100.0) | 5122 (100.0) |

Recent work on conceptualising and measuring psychological violence (here termed emotional violence) has suggested that it is distinct from controlling behaviour.[67] More than one-third of women in our study reported experiencing 1 of 12 controlling behaviours and other studies have reported high levels of control by intimate partners: 72% in rural Tamil Nadu[55] and 60% in rural Rajasthan,[56] 43% in urban Delhi[57] and 12% in Karnataka.[68] In a survey of married women in Uttar Pradesh, 12% had experienced reproductive control from husbands or in-laws, with a roughly equal split between them.[69]

Although emotional violence and coercive control are less visible than physical violence, the gravity of their effects on women's mental health is being increasingly appreciated.[70] Follingstad *et al* identified six types of emotional abuse that survivors of violence in the USA had experienced: verbal attacks (ridicule, name calling, humiliation in public), isolation (social or financial), jealousy and possessiveness (including accusations of infidelity), threats of harm, threats of divorce or abandonment and destruction of personal property. All had strong negative effects on women and 73% said that emotional violence had greater impact than physical violence.[71] This finding is supported by both qualitative[71 72] and quantitative research in high-income countries.[73] In an analysis of the US National Violence Against Women Survey, psychological violence was more strongly associated with depressive symptoms than physical violence, and abuse of power and control were more strongly associated with depressive symptoms than verbal abuse.[74]

We are seeing similar evidence from India. In rural Rajasthan, Richardson and colleagues '… found evidence that psychological abuse and controlling behavior were more damaging to mental health than physical abuse'.[56] A cross-sectional survey of 9938 mothers aged 15–49 years in seven urban and rural locations, of whom 3155 lived in urban informal settlements, suggested that severe harassment by in-laws was associated with poor mental health.[75] An analysis of follow-up data from 6303 rural married women aged 15–49 years in Bihar, Jharkhand, Maharashtra and Tamil Nadu showed a strong association between 'verbal' violence and mental health and a much weaker association with physical violence.[76]

### Generalisability
While domestic violence is a global experience, there are differences in cultural experiences. A feature of the discussion in high-income countries from the 1970s onward was the disappointment of professional providers of support when a survivor of intimate partner violence did not leave the perpetrator.[23 24 77] The emphasis in India—on the part of survivors and professionals—has been on interventions that allow families to remain together. Tolerating domestic violence has been a normalised element of

women's reproductive labour,[78 79] and the ability to bear it and still fulfil the expectations of a wife and mother has been seen as praiseworthy[80 81] and characteristic of the 'real' or 'good' woman.[82] Even when domestic violence spills out into community knowledge, women who break with this tradition risk social ostracism,[83] and their perceptions of their options are limited.[27 65]

## CONCLUSION

Domestic violence against women continues to be common in urban informal settlements in Mumbai and we see no reason to doubt the external validity of our findings for other locations. Physical and sexual violence are perpetrated mainly by intimate partners, but emotional violence against women is attributed equally to partners and marital family. More than one-third of women describe at least one of a limited set of controlling behaviours, perpetrated more by intimate partners, but often by marital family members.

We have two recommendations for research and action. First, social workers and healthcare providers should be aware of the importance of emotional violence and coercive control. Both are common and cause substantial suffering, particularly to women's mental health. These forms of violence need to be considered in interactions with women because of their subtlety and their intersection with poverty in urban informal settlements. Researchers should make sure that they are included in studies of domestic violence.

Second, violence is often perpetrated by family members other than intimate partners. Again, this is particularly true of emotional violence and coercive control. In interacting with clients, social workers and healthcare providers need to be aware that family members accompanying them might be involved in abuse, and studies should assume the possibility of other perpetrators.

**Acknowledgements** Our greatest thanks go to the women and community guardians who agreed to contribute to the study. We thank the team of investigators who collected data in challenging conditions, Miheeka Vast and Manju Singh for supervising the field investigation team, Unnati Machchhar and Shilpa Adelkar for their supervision of the intervention programme, Gauri Savkur for contributing to investigator training, Bhaskar Kakad for investigator support, Archana Bagra and Vibhavari Bali for financial and human resources management, and Vanessa D'Souza and Shanti Pantvaidya for leadership at SNEHA.

**Contributors** ND and DO conceived the study and acquired funding. ND, AG, SK, LG and DO developed the methodology. ND, SK and AG oversaw investigation. SK and DO curated the data. DO did the first analysis and visualisation and wrote the first draft. ND managed the project with support from SK. All authors critically reviewed the drafts of the manuscript and read and commented on the final version.

**Funding** Wellcome Trust (206417).

**Disclaimer** The funder had no role in study design, data collection and analysis, decision to publish, or preparation of the manuscript.

**Competing interests** None declared.

**Patient and public involvement** Patients and/or the public were involved in the design, or conduct, or reporting, or dissemination plans of this research. Refer to the Methods section for further details.

**Patient consent for publication** Not required.

**Ethics approval** The trial and associated data collection were approved by the Institutional Ethics Committee of Partners for Urban Knowledge, Action and Research (PUKAR) (25 December 2017) and the University College London Research Ethics Committee (3546/003; 27 September 2017). The TARA trial within which data collection took place is registered with the Controlled Trials Registry of India (CTRI/2018/02/012047; 21 February 2018) and with ISRCTN84502355 (22 February 2018: http://www.isrctn.com/ISRCTN84502355).

**Provenance and peer review** Not commissioned; externally peer reviewed.

**Data availability statement** Data are available in a public, open access repository. Data are available in Open Science Framework:Osrin, D. (2020, December 5). Prevalence of domestic violence against women in informal settlements in Mumbai, India: a cross-sectional survey. Retrieved from osf.io/8y4wj

**ORCID iDs**
Lu Gram http://orcid.org/0000-0002-3905-0465
David Osrin http://orcid.org/0000-0001-9691-9684

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
