## [Reviewer comments · BMJ Open]

ARTICLE DETAILS

TITLE (PROVISIONAL)	Prevalence of domestic violence against women in informal settlements in Mumbai, India: a cross-sectional survey
AUTHORS	Daruwalla, Nayreen; Kanougiya, Suman; Gupta, Apoorwa; Gram, Lu; Osrin, David

VERSION 1 – REVIEW

REVIEWER	Dr. Suresh Jungari Savitribai Phule Pune University, India
REVIEW RETURNED	27-Jul-2020

GENERAL COMMENTS	Thank you for allowing me to read this interesting and important manuscript entitled "Prevalence of domestic violence against women in informal settlements in Mumbai, India: a cross-sectional survey". The study will be an important contribution to the literature to understand the prevalence and factors affecting violence against women in slums. The manuscript is well written with clear objectives and robust statistical analysis. However, I have a few comments for authors to consider while revising the manuscript: 1. Abstract: Methods section is not clear and confusing. Write clearly about total samples covered and sampling process.2. Methods: In study setting section you mixed both setting and SNEHA organization details giving less details of study setting. I suggest, elaborate on study setting and explain separately about SNEHA.3. Design section is confusing with tapping together a lot of information about study sample size and other details. I suggest authors to have separate subheading for sample size, sampling process, data collection and data analysis.4. Data collection was done by trained (3 Months) female investigators. However, what other precautions had been taken while conducting the interviews such as privacy, where interviews conducted, what kind of provisions were made in case violence occurred, what kind of provisions were made in case violence occurred, what kind of provisions were made in case violence occurred, what kind of provisions were made in case violence occurred? Did any social worker or counselor accompany with data collectors?5. Discussion: Discussion section needs revision. In second paragraph authors discuss about limitation. I suggest having a separate section on study limitation before conclusion section.6. Fourth paragraph of discussion section saying page 11 line 46-48 stating "it is also possible that women answer questions differently when they are responding to a survey on violence than when the questions on violence are included in a broad survey like NFHS" I suggest authors to support this statement with evidence.7. Study should state some policy or programme implications of in the slums.
--

REVIEWER	Amira Shaheen
-----------------	---------------

	An-Najah National University Nablus Palestine
REVIEW RETURNED	01-Aug-2020

GENERAL COMMENTS	Overall; this is an important article in the field of GBV, shedding the light on special, and disadvantage population. Kindly find enclosed my comments; 1. Abstract; The authors indicated interviewing 100 women per 50 clusters, this would leave us with 5000 women being interviewed. It is unclear for me how the number become 5122. 2. Methods; Interviews were carried on household. As authors described, it is overcrowded residency, so how the safety of the women as well as researchers was granted 3. Results; Table 3; The total is misleading. May be it worth mentioning in the table title that (n=5122). What about if a repetitive act was answered by women, how authors dealt with this in the analysis 4. Discussion; - I believe this needs to be re-structured based on the journal requirement; Overview about the main results, comparison with other studies, strength and weaknesses of this study, conclusions, policy implications, and future research. - Cluster effect was not discussed. It is unclear for me whether the authors had account for it during the analysis. - Effect of inter-observer variation was not discussed. - The conclusion does not reflect on policy implication of the results
--

VERSION 1 – AUTHOR RESPONSE

Reviewer 1

Abstract: Methods section is not clear and confusing. Write clearly about total samples covered and sampling process.

Response

We have amended the abstract:

“We did a cross-sectional survey in two large informal settlements areas of the city, aiming to include approximately 5000 participants (~100 women aged 18-49 years in each of 50 clusters).”

Methods: In study setting section you mixed both setting and SNEHA organization details giving less details of study setting. I suggest, elaborate on study setting and explain separately about SNEHA.

Response

We have amended this by splitting the section into two and adding more information about informal settlements.

Design section is confusing with tapping together lot of information about study sample size and other details. I suggest authors to have separate subheading for sample size, sampling process, data collection and data analysis.

Response

We have added subheadings as suggested and rearranged this part of the Methods section.

Data collection was done by trained (3 Months) female investigators. However, what other precautions had taken while conducting the interviews such as privacy, where interviews conducted,

what kind of provisions were made in case violence survived women broken down with her past experiences? Did any social worker or counselor accompanied with data collectors.

Response

We have added more information to the section on data collection:

“Interviewers were all women and provided both time and sufficient information for women to consider whether to participate. They were supported by three field supervisors with direct linkage to counselling services, available by phone at any time. The interview team visited the local police station and social services to discuss their activities before starting in each area. Interviewers worked in groups of 7-8 in one cluster at a time, accompanied by a supervisor. A pair worked together in each lane and administered interviews in adjacent households.

To ensure privacy, interviews were arranged by advance appointment and avoided times when partners or children were likely to return from work or school. Women were interviewed at home or in a local community office if they preferred it. The interview began with general questions about demography, household residents, education, socioeconomic position, maternity, and health. If a family member, neighbour, or friend entered, the interviewer went back to asking questions about general health. If the person showed signs of staying, the interview was terminated and completed over up to three repeat visits. As a result of the gatekeeper consent process, community members were aware that interviewers would be visiting people in their area and this limited curiosity and intrusion.”

Discussion: Discussion section need revision. In second paragraph authors discussing about limitation. I suggest having separate section on study limitation before conclusion section.

Response

The STROBE guideline suggests placing limitations in the current position. We have added subheadings to structure the Discussion section according the STROBE guideline, with some additions: key results, limitations, (comparison with other studies), implications, generalisability. Fourth paragraph of discussion section saying page 11 line 46-48 stating “it is also possible that women answer questions differently when they are responding to a survey on violence than when the questions on violence are included in a broad survey like NFHS” I suggest authors to support this statement with evidence.

Response

The statement was based on the article by Saltzman, which reviews a number of sources of estimates of prevalence of intimate partner violence. We have made this clear by adding the following:

“In a consideration of methodological issues in research on intimate partner violence, Saltzman suggests that questionnaire context and cues given to the participant can influence response (Saltzman, 2004).”

Study should state some policy or programme implications of in the slums.

Response

We have rewritten the conclusion section:

“We have two recommendations for research and action. First, social workers and healthcare providers should be aware of the importance of emotional violence and coercive control. Both are common and cause substantial suffering, particularly to women’s mental health. These forms of violence need to be considered in interactions with women because of their subtlety and their intersection with poverty in urban informal settlements. Researchers should make sure that they are included in studies of domestic violence.

Second, violence is often perpetrated by family members other than intimate partners. Again, this is particularly true of emotional violence and coercive control. In interacting with clients, social workers and healthcare providers need to be aware that family members accompanying them might be involved in abuse, and studies should assume the possibility of other perpetrators.”

Reviewer: 2

Abstract; The authors indicated interviewing 100 women per 50 clusters, this would leave us with 5000 women being interviewed. It is unclear for me how the number become 5122.

Response

Interviewers aimed to achieve 100 interviews per cluster, giving a total of 5000. As is usual with field data collection, which was divided among 16 people, there was some variation in the numbers achieved. A median 101 interviews were achieved in each cluster (interquartile range 100-103, range 94-118). We have added this to the Results section.

Methods; Interviews were carried on household. As authors described, it is overcrowded residency, so how the safety of the women as well as researchers was granted

Response

We have added text on this to the section on data collection:

“Interviewers were all women and provided both time and sufficient information for women to consider whether to participate. They were supported by three field supervisors with direct linkage to counselling services, available by phone at any time. The interview team visited the local police station and social services to discuss their activities before starting in each area. Interviewers worked in groups of 7-8 in one cluster at a time, accompanied by a supervisor. A pair worked together in each lane and administered interviews in adjacent households.

To ensure privacy, interviews were arranged by advance appointment and avoided times when partners or children were likely to return from work or school. Women were interviewed at home or in a local community office if they preferred it. The interview began with general questions about demography, household residents, education, socioeconomic position, maternity, and health. If a family member, neighbour, or friend entered, the interviewer went back to asking questions about general health. If the person showed signs of staying, the interview was terminated and completed over up to three repeat visits. As a result of the gatekeeper consent process, community members were aware that interviewers would be visiting people in their area and this limited curiosity and intrusion.”

And in the section on data management:

“Mobile connectivity allowed immediate contact between interviewers and supervisors.”

And in the section on ethical considerations:

“We made provision for storage of participant information sheets on women’s behalf if they were concerned that the paperwork might be seen by others.”

“Participants were able to speak with counsellors immediately by phone. When a survivor was not ready to disclose violence, the interviewer provided her with information on available services and legal rights and gave her a small card that was easy to hide and listed essential contact numbers and addresses for 24-hour crisis support, medical emergencies, and the police. She took consent for any action from the participant herself.”

Results; Table 3; The total is misleading. May be it worth mentioning in the table title that (n=5122). What about if a repetitive act was answered by women, how authors dealt with this in the analysis

Response

We included the total at the foot of each column. We have always been advised (and advise others) that including a 100% cell in a table shows the reader clearly the direction in which to understand the percentages. We have changed ‘total’ to ‘all’ in all the relevant tables, and have added the number of respondents and their age group to all the relevant table and figure titles. The questions about domestic violence allowed for repeated acts and for acts under different categories of violence.

Discussion; I believe this needs to be re-structured based on the journal requirement; Overview about the main results, comparison with other studies, strength and weaknesses of this study, conclusions, policy implications, and future research.

Response

We have restructured the discussion section and added subheadings to make the flow clear.

Cluster effect was not discussed. It is unclear for me whether the authors had account for it during the analysis.

Response

The analysis used survey commands which accounted for cluster as the primary sampling unit and for strata describing broader settlements.

Effect of inter-observer variation was not discussed.

Response

We assessed inter-observer variation in supervisory meetings with interviewers. In line with good data monitoring practice, we examined variation in prevalence rates by interviewer and selected interviewers for field observation who showed early signs of deviation from the group average.

We have added to the data collection section:

“We examined variation in prevalence rates by interviewer and discussed performance in supervisory meetings. We selected for field observation interviewers who showed signs of deviation from the group average and provided feedback where necessary.”

The conclusion does not reflect on policy implication of the results

Response

We have added to the conclusion:

“We have two recommendations for research and action. First, social workers and healthcare providers should be aware of the importance of emotional violence and coercive control. Both are common and cause substantial suffering, particularly to women’s mental health. These forms of violence need to be considered in interactions with women because of their subtlety and their intersection with poverty in urban informal settlements. Researchers should make sure that they are included in studies of domestic violence.

Second, violence is often perpetrated by family members other than intimate partners. Again, this is particularly true of emotional violence and coercive control. In interacting with clients, social workers and healthcare providers need to be aware that family members accompanying them might be involved in abuse, and studies should assume the possibility of other perpetrators.”

VERSION 2 – REVIEW

REVIEWER	Dr. Suresh Jungari Savitribai Phule Pune University, India
REVIEW RETURNED	03-Sep-2020
GENERAL COMMENTS	I appreciate authors for making substantial changes to bring out clarity. The manuscript may be accepted for publication.

VERSION 2 – AUTHOR RESPONSE

Thank you for accepting the paper for publication.

As far as we can see, the reviewer did not suggest any revisions.